# Identification of Genes Involved in Fe–S Cluster Biosynthesis of Nitrogenase in *Paenibacillus polymyxa* WLY78

**DOI:** 10.3390/ijms22073771

**Published:** 2021-04-05

**Authors:** Qin Li, Yongbing Li, Xiaohan Li, Sanfeng Chen

**Affiliations:** State Key Laboratory of Agrobiotechnology and College of Biological Sciences, China Agricultural University, Beijing 100193, China; lqliqin1@126.com (Q.L.); ybli@cau.edu.cn (Y.L.); lixiaohan21@126.com (X.L.)

**Keywords:** *Paenibacillus polymyxa*, Fe–S cluster, nitrogenase, nitrogen fixation

## Abstract

NifS and NifU (encoded by *nifS* and *nifU*) are generally dedicated to biogenesis of the nitrogenase Fe–S cluster in diazotrophs. However, *nifS* and *nifU* are not found in N_2_-fixing *Paenibacillus* strains, and the mechanisms involved in Fe–S cluster biosynthesis of nitrogenase is not clear. Here, we found that the genome of *Paenibacillus polymyxa* WLY78 contains the complete *sufCDSUB* operon, a partial *sufC2D2B2* operon, a *nifS*-like gene, two *nifU*-like genes (*nfuA*-like and *yutI*), and two *iscS* genes. Deletion and complementation studies showed that the *sufC*, *sufD,* and *sufB* genes of the *sufCDSUB* operon, and *nifS*-like and *yutI* genes were involved in the Fe–S cluster biosynthesis of nitrogenase. Heterologous complementation studies demonstrated that the *nifS*-like gene of *P. polymyxa* WLY78 is interchangeable with *Klebsiella oxytoca nifS*, but *P. polymyxa* WLY78 SufCDB cannot be functionally replaced by *K. oxytoca* NifU. In addition, *K. oxytoca nifU* and *Escherichia coli nfuA* are able to complement the *P. polymyxa* WLY78 *yutI* mutant. Our findings thus indicate that the NifS-like and SufCDB proteins are the specific sulfur donor and the molecular scaffold, respectively, for the Fe–S cluster formation of nitrogenase in *P. polymyxa* WLY78. YutI can be an Fe–S cluster carrier involved in nitrogenase maturation in *P. polymyxa* WLY78.

## 1. Introduction

Iron–sulfur (Fe–S) clusters are contained in a diverse group of proteins called Fe–S proteins, which participate in a wide variety of cellular processes, such as nitrogen fixation, respiration, DNA repair, and gene regulation [1,2,3,4]. So far, three pathways for Fe–S cluster assembly identified in bacteria are the *ni*trogen *f*ixation (Nif) system, the *i*ron *s*ulfur *c*luster (Isc) system, and the *su*lfur *f*ormation (Suf) system [5]. The Nif system was initially discovered in the maturation of nitrogenase in *Azotobacter vinelandii* [6]. Nitrogenase that catalyzes biological nitrogen fixation comprises two components, the Fe protein and the MoFe protein, and both of these are Fe–S proteins. Genetic and biochemical analysis has revealed that the products of *nifU* and *nifS* are specifically required for activation of both Fe and MoFe proteins [7,8]. Subsequent studies have suggested that NifS is a cysteine desulfurase that catalyzes the production of sulfur from L-cysteine, while NifU provides a molecular scaffold for the assembly and transfer of the Fe–S cluster to the components of nitrogenase [9,10,11].

In addition to the Nif system that is specifically responsible for the synthesis of the Fe–S clusters of nitrogenase, Isc and Suf are required for a broad range of cell functions [5]. *Escherichia coli* and closely related enterobacteria possess both Isc and Suf systems, encoded by the *iscRSUA-hscBA-fdx-iscX* operon and the *sufABCDSE* operon, respectively. The Isc system functions under normal growth conditions, whereas the Suf system operates under stress conditions such as iron starvation and oxygen limitation [12,13,14]. In the Isc machinery, IscS is a cysteine desulfurase that serves as a sulfur donor and IscU is a scaffold protein. HscA/HscB (molecular chaperones) and ferredoxin (Fdx) interact with IscU containing the Fe–S cluster and facilitate the cluster transfer from IscU to target proteins. IscA plays a role in the transfer of the Fe–S cluster from IscU to target proteins, while IscR is a negative regulator [15,16,17]. In the Suf system, SufS is a cysteine desulfurase and SufE serves as a sulfur shuttle protein that interacts with SufS. The three proteins SufB, SufC, and SufD associate as a SufBCD complex that functions as a scaffold. Thus, SufSE produces sulfur and SufCDB as a scaffold assembles an Fe–S cluster. SufA is an IscA homologue and acts as an Fe–S cluster-specific carrier to target proteins [18,19,20,21]. 

In the Gram-positive model of the bacterium *Bacillus subtilis*, Fe–S cluster assembly is mediated by the Suf system consisting of a *sufCDSUB* operon and a distant *sufA* [22]. *B. subtilis* SufS functions as a cysteine desulfurase, and SufBCD is a scaffold, both of which are functionally similar to the corresponding *E. coli* Suf components [23]. *B. subtilis* SufU interacts with SufS and is involved in sulfur transfer from SufS to SufBCD, just as SufE is involved in the *E. coli* Suf system. *B. subtilis* SufSU is interchangeable with *E. coli* SufSE but not with IscSU [20,24].

In addition to the three distinct Nif, Suf, and Isc systems, some Fe–S cluster biosynthetic genes have been identified. For example, *E. coli* NfuA binds a 4Fe–4S cluster and transfers this cluster to the target protein and acts as a scaffold/chaperone for damaged Fe–S proteins under oxidative stress and iron starvation conditions [25,26,27]. *E. coli* CsdA is the third cysteine desulfurase except for IscS and SufS, and it is also engaged in two separate sulfur transfer pathways by recruiting Suf components and by interacting with CsdE and CsdL participates [28].

*Paenibacillus polymyxa* WLY78 is a Gram-positive, facultative anaerobic, endospore-forming, N_2_-fixing bacterium. The genome of this bacterium contains a minimal and compact *nif* gene cluster composed of nine genes *(nifB nifH nifD nifK nifE nifN nifX hesA nifV*) [29]. This minimal *nif* cluster has great potential use in engineering nitrogen fixation into non-N_2_-fixing organisms [30]. Our recent study demonstrated that the minimal *nif* gene cluster enables *E. coli* to synthesize the catalytically active nitrogenase, but the specific activity of the enzyme expressed in *E. coli* was only approximately 10% of that observed in *Paenibacillus* [29]. Compared to the *nif* gene clusters of *A. vinelandii* and *Klebsiella oxytoca,* the *P. polymyxa* WLY78 *nif* cluster lacks *nifS* and *nifU*. Thus, we attempted to identify whether *nifS-*like and *nifU-*like or other Fe–S cluster biosynthetic genes are responsible for the Fe–S cluster assembly of nitrogenase.

Here, we searched the genome of *P*. *polymyxa* WLY78 and found that there are an entire *sufCDSUB* operon, a partial *sufC2D2B2* operon, a partial *isc* system (*iscSR*), and a single *iscS* (*iscS2*)*,* one *nifS*-like, and two *nifU*-like genes (*nfuA*-like and *yutI*). Mutation and complementation analysis of these putative Fe–S assembly genes revealed that SufCDB and NifS-like proteins are essential for biosynthesis of nitrogenase. Furthermore, heterologous complementation studies demonstrated that *K. oxytoca nifS* is able to restore the nitrogenase activity of the *nifS*-like mutant of *P. polymyxa* WLY78, but *K. oxytoca nifU* cannot complement any of the *sufC*, *sufD*, and *sufB* mutants. We also demonstrated that *yutI* is involved in nitrogen fixation, but other genes are not. Our study not only reveals the distribution and functions of the Fe–S cluster biosynthetic genes involved in nitrogen fixation in *P. polymyxa* WLY78, but also provides insight into the mechanisms of Fe–S cluster assembly.

## 2. Results

### 2.1. The Putative Fe–S Cluster Biosynthetic Genes in the Genome of P. polymyxa WLY78

We searched the genome of *P. polymyxa* WLY78 by using MetalPredator (http://metalweb.cerm.unifi.it/tools/metalpredator/, accessed on 23 February 2020) and identified 118 genes encoding putative Fe–S cluster-containing proteins and Fe–S cluster biosynthetic genes (Appendix A). As shown in Figure 1, the Fe–S cluster biosynthetic genes in *P. polymyxa* WLY78 include an entire Suf system (*sufCDSUB*), a *sufA* gene, a partial Suf system (*sufC2D2B2*), a partial Isc system (*iscSR*), a single *iscS2* gene, and a *nifS*-like and two *nifU*-like (*nfuA*-like and *yutI*) genes.

In the genome of *P. polymyxa* WLY78, *sufC*, *sufD*, *sufS*, *sufU,* and *sufB* (*sufCDSUB*) are tightly arranged with the same transcriptional direction. However, *sufA* is not in the *sufCDSUB* cluster of *P. polymyxa* WLY78, which differs from the *suf* operon arrangement in *E. coli*. The current analysis using BLAST alignment showed that the SufA protein of *P. polymyxa* WLY78 shares 26.23% and 48.33% identities with *E. coli* and *B. subtilis* SufA proteins, respectively. The complete *sufCDSUB* operon of *P. polymyxa* WLY78 shows a similar arrangement as that of *B. subtilis.* The SufCDSUB proteins showed 18.11–46.96% identity with their corresponding components from *E. coli* Suf and 47.84–76.82% identity with their corresponding components from *B. subtilis* Suf at the amino acid level (Figure 2A). In addition, *P. polymyxa* WLY78 carries a partial *suf* operon (here designated as *sufC2D2B2*) whose predicted products SufC2, SufD2, and SufB2 showed 84.88%, 62.27%, and 84.73% identities with their corresponding components of the entire Suf system, respectively.

The genome of *P. polymyxa* WLY78 contains an *iscR* gene and two *iscS* genes. Of the two *iscS* genes, one is linked to *iscR* as a dicistronic *iscSR* operon and the other (here designated as *iscS2*) is located elsewhere. The *iscRS* operon of *P. polymyxa* WLY78 has a similar organization as that of *B. subtilis*. However, compared to the *isc* operon (*iscRSUA-hscBA-fdx*) of *A. vinelandii* and *E. coli*, the *P. polymyxa* WLY78 *isc* operon (*iscRS*) lacks *iscUA* and other genes (Figure 2B).

Unlike *A. vinelandii* and *K. oxytoca*, *P. polymyxa* WLY78 does not have *nifS* and *nifU* but contains a *nifS*-like gene and two *nifU*-like genes (*nfuA*-like and *yutI*) that are not associated with the *nif* gene cluster (*nifB nifH nifD nifK nifE nifN nifX hesA nifV*). *P. polymyxa* WLY78 has a *nifS*-like gene encoding a 397-amino-acid protein, which showed 28.77% and 28.36% identity with NifS of *A. vinelandii* and *K. oxytoca*, respectively (Figure 3). Similar to *A. vinelandii* and *K. oxytoca* NifS, the NifS-like protein of *P. polymyxa* WLY78 has conserved residues that may be involved in pyridoxal-phosphate (PLP) binding and invariable residues Cys that may be involved in substrate binding. A protein alignment revealed that the amino acid sequences derived from two NifU-like proteins with a single domain (NfuA-like and YutI) have high similarity with those of the C-terminal domain of NifU from *A. vinelandii* and *K. oxytoca* and the C-terminal domain of NfuA from *E. coli* and *A. vinelandii* (Figure 4). Moreover, they have a strict conservation of the CXXC motif, which is the predicted site for Fe–S cluster assembly.

### 2.2. NifS-Like Protein Is Essential for Nitrogenase Synthesis

To determine whether the NifS-like protein is involved in the Fe–S cluster assembly of nitrogenase in *P. polymyxa* WLY78, we constructed an in-frame deletion mutant (Δ*nifS*-like) and a complementation strain (Δ*nifS*-like/*nifS*-like). In comparison with wild-type *P. polymyxa* WLY78, the Δ*nifS*-like mutant exhibited nearly no activity under the nitrogen-limited condition, indicating that *nifS*-like is required for nitrogen fixation. Complementation of Δ*nifS*-like with the *nifS*-like gene carried in the plasmid was able to restore nitrogenase activity, suggesting that a change in nitrogenase activity was due solely to the deletion of *nifS*-like (Figure 5A). Moreover, heterologous complementation of the Δ*nifS*-like mutant with *K. oxytoca nifS* partially restored the effect of *nifS*-like mutation. These data suggest that the *nifS*-like gene of *P. polymyxa* WLY78 and the *nifS* gene of *K. oxytoca* are similar in function. 

### 2.3. NifU-Like Protein (YutI) Is Involved in Nitrogen Fixation, but NfuA-Like Protein Is Not

Sequence analysis indicates the presence of two genes (*nfuA*-like and *yutI* genes), encoding for 85 and 81 amino acids, respectively, that share similarity with the C-terminus of the Fe–S scaffold protein NifU and the Fe–S carrier NfuA in *A. vinelandii* (Figure 4). The current analysis using the BLAST alignment showed that NfuA-like and YutI proteins encoded by *nfuA*-like and *yutI* genes have 61.73% identity. In this study, the in-frame deletion mutants Δ*nfuA*-like and Δ*yutI* were constructed. The Δ*nfuA*-like mutant showed almost similar nitrogenase activity as wild-type *P. polymyxa* WLY78, suggesting that the *nfuA*-like gene is not involved in nitrogen fixation (Figure 5B).

The activity of the Δ*yutI* mutant was approximately 50% that in wild-type *P. polymyxa* WLY78 (Figure 5B). Complementation experiments showed that the *yutI* gene carried on the plasmid was able to restore the nitrogenase activity of the Δ*yutI* mutant. The data suggest that *P. polymyxa* WLY78 *yutI* is involved in nitrogen fixation. Deletion of both *nfuA* and *yutI* (Δ*nfuA*-like/Δ*yutI* double mutant) did not result in a further decline in nitrogenase activity, suggesting that NfuA-like has a limited ability to serve the function of YutI in nitrogen fixation. Furthermore, heterologous complementation studies showed that *K. oxytoca nifU* and the *E. coli nfuA* gene were able to restore the nitrogenase activity of the Δ*yutI* mutant, suggesting that *P. polymyxa yutI* performs similar functions as *K. oxytoca nifU* and *E. coli nfuA*.

To investigate whether the two conserved cysteine residues (Cys-49 and Cys-52) in the YutI protein play roles in Fe–S assembly, the Δ*yutI* mutant was complemented with the mutated *yutI* gene whose coding product Cys-49 or Cys-52 or both cysteine residues were replaced with alanine residue(s). As shown in Figure 5B, the YutI protein with mutated Cys-49 or Cys-52, especially with the two mutated cysteine residues, could not complement the *yutI* mutant as well as the wild-type *yutI* gene did. These results indicate that the conserved Cys-49 and Cys-52 in the YutI protein are responsible for the Fe–S cluster binding of nitrogenase.

### 2.4. IscS Is Not Required for Nitrogen Fixation

Although *P. polymyxa* WLY78 contains two *iscS* genes (*iscS* and *iscS2*), no IscU homologues have been identified. Amino acid sequence analysis reveals that IscS and IscS2 of *P. polymyxa* have 35% identity. However, only IscS has conserved the SSGSACTS sequence, which represents the feature of the IscS protein (Appendix A). We constructed in-frame deletion mutants Δ*iscS* and Δ*iscS2* and the double mutant Δ*iscS/*Δ*iscS2* and found that nitrogenase activities were similar in all *iscS* mutants and the wild-type strain under the N_2_-fixing condition (Figure 6A), suggesting that both IscS and IscS2 proteins are not required for nitrogen fixation in *P. polymyxa* WLY78.

### 2.5. The sufCDSUB Operon Is Required for Nitrogenase, but the sufB2C2D2 Operon and sufA Gene Is Not

In *B. subtilis*, there is only the Suf pathway for Fe–S generation and the knockout of *suf* genes (any one of *sufCDSUB*) is lethal [23]. However, *P. polymyxa* WLY78 contains another partial *suf* operon (*sufC2D2B2*) due to which the inactivation of *sufCDB* does not lead to lethality. Attempts to knock out *sufSU* in *P. polymyxa* WLY78 were unsuccessful, suggesting that *sufSU* might exist to maintain basic levels of Fe–S assembly for survival. In this study, the deletion mutants, including Δ*sufC*, Δ*sufD*, Δ*sufB*, Δ*sufCDB*, Δ*sufC2D2B2,* and Δ*sufA*, were successfully disrupted. qRT-PCR further revealed that the messenger RNA (mRNA) levels of other *suf* genes in the *sufCDSUB* operon were not affected by one *suf* gene mutation (Appendix A), suggesting that the deletion did not generate a polarity effect.

The nitrogenase activities of these *suf* mutants and wild-type *P. polymyxa* WLY78 were comparatively analyzed (Figure 6B). The *suf* gene deletion mutants (Δ*sufC*, Δ*sufD*, Δ*sufB,* and Δ*sufCDB*) showed a significant decrease in nitrogenase activity compared to the wild-type strain, while Δ*sufB2C2D2* and Δ*sufA* mutants exhibited almost similar activity as the wild-type strain. Furthermore, complementation of Δ*sufC*, Δ*sufD,* and Δ*sufB* mutants with the corresponding *P. polymyxa* WLY78 *suf* gene restored nitrogenase activity to the wild-type level, suggesting that *sufCDB* genes are essential for the Fe–S cluster assembly of nitrogenase in *P. polymyxa* WLY78. We also tried to complement with *K. oxytoca nifU* and found that *nifU* could not restore the nitrogenase activity of any *suf* mutants (Figure 6B). These results indicate that *K. oxytoca* NifU cannot replace SufCDB in *P. polymyxa* WLY78.

## 3. Discussion

Nitrogenase is a complex metalloenzyme, and the Fe–S clusters of nitrogenase play a critical function in electron transfer and in the reduction of substrates driven by the free energy liberated from Mg-ATP hydrolysis [6,11,31]. In the N_2_-fixing model bacteria *A. vinelandii* and *K. oxytoca*, NifS and NifU are specifically responsible for the Fe–S cluster assembly of nitrogenase, and *nifU* and *nifS* usually are clustered with other *nif* genes [32,33]. However, the *nif* clusters of *P*. *polymyxa* WLY78 and other N_2_-fixing *Paenibacillus* species do not have *nifS* and *nifU* [34]. In this study, we report that *nifS*-like, *nifU*-like (*yutI*), and *sufCDB* genes are involved in the Fe–S cluster assembly of nitrogenase in *P*. *polymyxa* WLY78.

The NifS protein was involved in providing sulfur to the iron–sulfur clusters of nitrogenase [35]. In this study, we revealed that the *nifS*-like gene of *P*. *polymyxa* WLY78 is involved in nitrogen fixation. The NifS-like protein from *P*. *polymyxa* WLY78 and NifS from *K. oxytoca* and *A. vinelandii* have common characteristics, which contain conserved residues known to be essential for activity, substrate recognition, and PLP binding. However, *P*. *polymyxa* WLY78 *nifS*-like exhibits two features different from *A.* v*inelandii nifS* and *K. oxytoca nifS*. One is that each of *A.* v*inelandii nifS* and *K. oxytoca nifS* is linked together with *nifU* as a *nifSU* operon that is located in a large *nif* gene cluster [32,33], while *P*. *polymyxa* WLY78 *nifS*-like is not linked together with any *nif* genes. The other feature is that there is a C-terminal extension consisting of 20-21 amino acids, including the consensus sequence SPL(W/Y)(E/D)(M/L)*X*(K/Q)*X*G(I/V)D(L/I)*XX*(/V)*X*W*XXX* in *A.* v*inelandii* NifS and *K. oxytoca* NifS [36], while this extension is absent in the NifS-like protein of *P*. *polymyxa* WLY78. Deletion of the *A. vinelandii nifS* gene did not lead to complete loss of nitrogenase activity, suggesting that the housekeeping Fe–S cluster biosynthetic system Isc could weakly replace NifS function [6,37]. We deduce that NifS activity could be replaced at low levels by the housekeeping Fe–S protein SufSU in *P*. *polymyxa* WLY78, since loss of NifS-like function does not completely eliminate the capacity for nitrogen fixation. *K. oxytoca nifS* could complement the *P*. *polymyxa* WLY78 *nifS*-like mutant, which further confirmed that the NifS-like protein is the specific S donor for the assembly of Fe–S clusters for nitrogenase in *P*. *polymyxa* WLY78.

NifU was shown as an Fe–S scaffold protein constructed of several domains for maturation of the nitrogenase [38]. The NfuA protein (known as NifU-like proteins) with sequence similarity to the C-terminal domain of NifU was found to be highly conserved among different bacteria [27]. *P*. *polymyxa* WLY78 possess two NifU proteins (NfuA-like and YutI) that share significant sequence homology to the C-terminus domain of the NifU family and include the conserved CXXC motif. Such sequence conservation indicates a likely role for NfuA-like and YutI in Fe–S protein maturation in *P*. *polymyxa* WLY78. In *A.* v*inelandii*, the NfuA protein contains the N-terminal A-type domain and the C-terminal Nfu-type domain, and functional loss of the Nfu-type domain has no obvious effect on nitrogenase maturation [39]. In this study, YutI has only one Nfu domain, which was found to be required for nitrogen fixation in *P*. *polymyxa* WLY78, and deletion of *yutI* results in lower nitrogenase activity. The conserved cysteine residues in the CXXC motif were shown to be functionally important for NfuA in *E. coli*, *A. vinelandii*, and *Pseudomonas aeruginosa* [26,39,40,41]. Accordingly, our results demonstrate that the Cys-49 and Cys-52 residues of the CXXC motif are essential for YutI to carry out its fixating nitrogen function in vivo. Previous studies have shown that the elevated expression of NifU could replace the function of NfuA in *A. vinelandii* [39]. The heterologous complementation assays performed in this study provide compelling evidence that the NifU of *K. oxytoca* and the NfuA of *E. coli* are functionally interchangeable with the YutI of *P*. *polymyxa* WLY78. The NfuA as an Fe–S cluster carrier was able to accept the Fe–S cluster from scaffold proteins IscU or SufBCD in vitro [27]. Therefore, the YutI of *P*. *polymyxa* WLY78 could potentially be a stand-alone carrier protein, playing a role similar to NfuA to accept the Fe–S cluster from the scaffold protein, as described in other microorganisms.

The cysteine desulfurase IscS is a highly conserved and essential component of the Isc system that serves as a sulfur donor for Fe–S cluster biogenesis [16]. In the present work, we showed that the *iscS* and *iscS2* of *P*. *polymyxa* WLY78 are not required for nitrogen fixation. The result is in line with our previous work that showed that the *iscSR* system cannot increase any activity of *E. coli* 78-7 carrying a *Paenibacillus nif* gene operon [42]. In addition, the Isc system is used for the maturation of other Fe–S proteins but not nitrogenase in *A. vinelandii* [6,43]. Inactivation of *iscS* led to no defects in Fe–S metabolism, suggesting that IscS is not used for housekeeping Fe–S protein maturation in *P*. *polymyxa* WLY78. However, in *B. subtilis*, the *iscS* gene could not be deleted, indicating that IscS participates in essential cellular processes [44].

Besides the NifSU-like and Isc system, the Suf system (the *sufCDSUB* operon) is highly conserved in N_2_-fixing *Paenibacillus* strains [34]. The Suf system has been characterized in several organisms such as *E. coli* [14,45], *B. subtilis* [23], *Staphylococcus aureus* [46], and *Synechocystis* [47]. Mutations in the *suf* operon can have severe consequences: for instance, disruption of *sufCDSUB* or its individual gene is lethal for *B. subtilis* [23]. The *P*. *polymyxa suf* genes had not been characterized before; therefore, it was not clear what impact the disruption of *suf* would have on the metabolism and viability. Directly obtaining the *sufCDSUB* operon mutant of *P*. *polymyxa* WLY78 was unsuccessful, which places the Suf component as key proteins in Fe–S cluster metabolism. This is in accordance with the fact the Suf Fe–S cluster biosynthetic system is essential for Gram-positive bacteria viability. However, in this study, we disrupted single *suf*C, *sufD,* and *sufB* genes of the *suf* operon successfully, showing that they are required for nitrogen fixation. This is in line with our previous work that the transcript abundances of *sufCDSUB* was up-regulated in N_2_-fixing condition [48] and the *suf* (*sufCDSUB)* operon can increase nitrogenase activity of *E. coli* 78-7 from 10% to 20% [42]. *suf*S and *sufU* could not be deleted, suggesting a critical role of these proteins for the survival of *P*. *polymyxa* WLY78. Our finding that the *sufCDB* mutant phenotype could not be rescued by the *nifU* of *K. oxytoca* demonstrated that SufCDB is not interchangeable with the NifU of *P*. *polymyxa* WLY78. RT-PCR analysis indicated that *nifU* was transcribed in these complementation strains (Appendix A). We considered one possible explanation for the inability of NifU to replace functionally SufCDB. Namely, that NifU does not productively interact with SufSU and that such a specific interaction might be required for maturation of Fe–S proteins. Taken together, we conclude that *P*. *polymyxa* WLY78 *sufCDB* plays an important role in the Fe–S cluster assembly of nitrogenase.

Deletion of *nifS*-like, *yutI,* or *sufBCD* genes leads to a decrease in nitrogenase activity. One reason is insufficiency to supply the Fe–S clusters necessary for nitrogenase maturation. It is also possible that the effect of deletion on the synthesis of nitrogenase might be direct or indirect by affecting *nif* transcription and Nif expression, as described in *K. oxytoca* [49]. qRT-PCR revealed that the transcription levels of *nifH* and *nifD* in these mutants exhibited a 2–5-fold decrease compared to the wild-type strain (Appendix A). Western blot analysis with the protein extracts also demonstrated that mutations affected the amounts of both the Fe protein (NifH) and the MoFe protein (NifD) (Appendix A). According to our results, we proposed the mechanisms involved in the Fe–S cluster biosynthesis of nitrogenase in *P*. *polymyxa* WLY78 (Figure 7). The Fe–S cluster assembles on a scaffold protein SufCDB, which receives sulfur from a cysteine desulfurase NifS-like and iron from an as yet unidentified source. Then, the pre-formed Fe–S cluster is transferred to a carrier protein YutI, which delivers it to the final apo-nitrogenase. Further studies are required to determine how these Fe–S proteins are involved in the formation and maturation of nitrogenase. Homologues of NifS and NifU have been identified in non-nitrogen-fixing organisms such as *E. coli*, yeast, and plants, leading to the proposal that attempts to engineer eukaryotic species for heterologous nitrogen fixation activity can incorporate an endogenous native Fe–S cluster.

## 4. Materials and Methods

### 4.1. Strains, Plasmids, and Growth Conditions

Bacterial strains and plasmids used in this study are summarized in Appendix A. *P*. *polymyxa* strains and *E. coli* strains were routinely grown in LB (per liter contains 10 g of NaCl, 5 g of yeast, and 10 g of tryptone) or LD medium (per liter contains 5 g of NaCl, 5 g of yeast, and 10 g of tryptone) at 30 °C with shaking. For assays of nitrogenase activity, *P*. *polymyxa* strains were grown in nitrogen-limited media under anaerobic conditions. The nitrogen-limited media contained (per liter) 0.4 g of Na_2_HPO_4_, 3.4 g of KH_2_PO_4_, 26 mg of CaCl_2_·2H_2_O, 30 mg of MgSO_4_, 0.3 mg of MnSO_4_, 36 mg of ferric citrate, 7.6 mg of Na_2_MoO_4_·2H_2_O, 10 mg of p-aminobenzoic acid, 5 µg of biotin, 2 mM glutamate, and 4 g of glucose as the carbon source. *E. coli* strains JM109 were used for routine cloning. Thermo-sensitive vector pRN5101 [50] was used for gene disruption in *P*. *polymyxa* WLY78. The shuttle vector pHY300PLK was used for the complementation experiment. When appropriate, antibiotics were added in the following concentrations for maintenance of plasmids: 100 μg/mL of ampicillin, 12.5 μg/mL of tetracycline, and 5 μg/mL of erythromycin.

### 4.2. Construction of ΔnifS-Like, ΔnfuA-Like, ΔyutI, ΔnfuA-Like/ΔyutI, ΔiscS, ΔiscS2, ΔiscS/ΔiscS2, ΔsufC, ΔsufD, ΔsufB, ΔsufCDB, ΔsufC2D2B2, and ΔsufA Mutants 

The in-frame-deletion mutants Δ*nifS*-like*,* Δ*nfuA*-like, Δ*yutI*, Δ*nfuA*-like/Δ*yutI*, Δ*iscS*, Δ*iscS2*, Δ*sufC*, Δ*sufD*, Δ*sufB,* Δ*sufA,* and Δ*sufC2D2B2* were constructed through homologous recombination with pRN5101. The upstream (ca. 1 kb) and downstream fragments (ca.1.0 kb) flanking the coding region of *iscS*, *iscS2*, *nifS-*like*, nfuA-*like, *yutI*, *sufC*, *sufD*, *sufB, sufCD*, *sufA,* and *sufC2D2B2* were separately PCR-amplified from the genomic DNA of *P*. *polymyxa* WLY78. The primers used for these PCR amplifications are listed in Appendix A. The two fragments flanking each coding region of *iscS*, *iscS2*, *nifS-*like*, nfuA-*like, *yutI*, *sufC*, *sufD*, *sufB, sufCD*, *sufA,* and *sufC2D2B2* were then fused with the *Hind* III/*BamH* I-digested pRN5101 vector using the Gibson assembly master mix (New England Biolabs, Ipswich, USA), generating the recombinant plasmids pRDiscS, pRDiscS2, pRDnifS, pRDnfuA, pRDyutI, pRDsufC, pRDsufD, pRDsufB, pRDsufCD, pRDsufA, and pRD*sufC2D2B2*. Then, each of these recombinant plasmids was transformed into *P*. *polymyxa* WLY78, and the double-crossover transformants Δ*iscS*, Δ*iscS2*, Δ*nifS-*like*,* Δ*nfuA-*like, Δ*yutI*, Δ*sufC*, Δ*sufD*, Δ*sufB,* Δ*sufCD*, Δ*sufA,* and Δ*sufC2D2B2* were selected from the initial Em^r^ transformants after several rounds of nonselective growth at 39 °C and confirmed by PCR amplification and sequencing analysis. Further, pRD*nfuA*, pRDiscS2, and pRDsufCD were transformed into Δ*nfuA*-like, Δ*iscS,* and Δ*sufB*, respectively, to generate double mutants Δ*nfuA*-like/Δ*yutI*, Δ*iscS*/Δ*iscS2,* and Δ*sufCDB*.

### 4.3. Construction of Plasmids for Complementation of P. polymyxa WLY78 Mutants

Complementation of Δ*nifS*-like*,* Δ*yutI*, Δ*sufC*, Δ*sufD*, Δ*sufB,* and Δ*sufCDB* was performed. For complementation of Δ*nifS*-like, a 1578 bp DNA fragment containing the coding region of *nifS-*like and its own promoter was PCR-amplified from the genomic DNA of *P*. *polymyxa* WLY78. For complementation of the Δ*yutI* mutant, a 626 bp DNA fragment containing the coding region of *yutI* and its own promoter was PCR-amplified. For complementation of the Δ*sufC* mutant, a 1173 bp DNA fragment carrying the *sufC* coding region and its own promoter was PCR-amplified. For complementation of Δ*sufD* and Δ*sufB* mutants, a 390 bp promoter region of the *sufCDSUB* operon and their respective coding region were PCR-amplified. For complementation of the Δ*sufCDB* mutant, a 2501 bp DNA fragment containing the coding region of *sufCD* and its promoter, and a 1398 bp *nifB* coding sequence was PCR-amplified. These fragments were digested with *BamH*I/*Hind*III and ligated into the vector pHY300PLK, generating vectors pHYnifS, pHYyutI, pHYsufC, pHYsufD, pHYsufB, and pHYsufCDB. Each of these recombinant plasmids was correspondingly transformed into its mutants, and tetracycline-resistant (Tc^r^) transformants were selected and confirmed by PCR and sequencing. 

The substituted forms of YutI were constructed by the PCR-based mutagenesis method. The pairs of mutagenic primer were designed to be complementary to each other and to span the substituted site. To produce the YutIC49A variant, two separate PCR reactions were conducted with *P*. *polymyxa* WLY78 as the template and by using the primer sets P-yutI-F/YutIC49A-R and YutIC49A-F/C-yutI-R. The two PCR products were then fused with the *Hind* III/*BamH*I-digested pHY300PLK vector, yielding the complementary plasmid pHY*yutI-*C49A, and then the plasmid was transformed into Δ*yutI*. The same procedure was to obtain pHY*yutI-*C52A and pHY*yutI-*C49/52A. The accuracy of mutagenesis was verified by DNA sequencing.

For heterologous complementation assays, the *nifS* gene was PCR-amplified from the genomic DNA of *K. oxytoca* M5a1, being under the control of the *nifS*-like promoter of *P. polymyxa* WLY78, and cloned to the plasmid pHY300PLK, generating the vector pHYnifS (*K.o*). This vector was transformed into a Δ*nifS-*like mutant, yielding the heterologous complementation strain Δ*nifS-*like*/nifS-K.o.* An 829 bp DNA fragment containing the coding region of *nifU* from *K. oxytoca* M5a1 and a 365 bp promoter region of *yutI* were PCR-amplified and assembled, generating the vector pHYnifU. Then, the constructed vector pHYnifU was transformed into the Δ*yutI*, Δ*sufC*, Δ*sufD*, Δ*sufB,* and Δ*sufCDB* mutants, yielding the heterologous complementation strain Δ*yutI/nifU-K.o*, Δ*sufC/nifU-K.o*, Δ*sufD/nifU-K.o*, Δ*sufB/nifU-K.o,* and Δ*sufCDB/nifU-K.o*, respectively. Similarly, a 591 bp DNA fragment containing the coding region of *nfuA* from *E. coli* and a 365 bp promoter region of *yutI* were PCR-amplified and assembled, generating the vector pHYnfuA (*E. coli*). Then it was transformed into the Δ*yutI* mutant, yielding the heterologous complementation strain Δ*yutI/nfuA-E. coli.*

### 4.4. Acetylene Reduction Assays of Nitrogenase Activity

Acetylene reduction assays were performed, as described previously, to measure nitrogenase activity [29]. *P. polymyxa* WLY78 and its mutant strains were grown overnight in LD medium. The cultures were collected by centrifugation, washed three times with sterilized water, and then resuspended in a nitrogen-limited medium containing 2 mM glutamate as a nitrogen source to a final OD_600_ of 0.2–0.4. Then, 4 mL of the culture was transferred to a 25 mL test tube, and the test tube was sealed with a rubber stopper. The headspace (21 mL) in the tube was then evacuated and replaced with argon gas. Then, 2.1 mL of C_2_H_2_ (10% of the headspace volume) was injected into the test tube. Cultures were incubated at 30 °C. C_2_H_4_ production was analyzed by gas chromatography. The nitrogenase activity was expressed in nmol C_2_H_4_/mg protein/h. The nitrogenase activity assays were measured at least three times, and the error bars were evaluated as the standard deviation. Statistical analyses were performed using SPSS software version 20 (SPSS Inc., Chicago, IL, USA). One-way ANOVA was employed to check the significant differences between strains. Means of different strains were compared using the least significant difference (LSD) at the 0.05, 0.01, or 0.001 level of probability.

### 4.5. qRT-PCR and RT-PCR Analysis of Gene Expression

For quantitative real-time-PCR (qRT-PCR), cultures of *P. polymyxa* WLY78, ∆*nifS-*like, ∆*yutI*, ∆*sufCDB,* ∆*sufC,* and ∆*sufD* were grown under N_2_-fixing conditions (2 mM glutamate and without O_2_) and harvested after 8 h of incubation. To detect the expression of the *nifU* gene, total RNA was extracted from ∆*sufC,* ∆*sufD,* ∆*sufB,* ∆*sufCDB,* ∆*sufC/nifU-K.o*, ∆*sufD/nifU-K.o*, ∆*sufB/nifU-K.o,* and ∆*sufCDB/nifU-K.o*. Total RNA was isolated using TRIzol (Takara Bio, Tokyo, Japan). The possibility of contamination of genomic DNA was eliminated by digestion with RNase-free DNase I (Takara Bio, Tokyo, Japan). The integrity and size distribution of the RNA were verified by agarose gel electrophoresis, and the concentrations were determined spectrophotometrically. Synthesis of cDNA was carried out using RT Prime Mix according to the manufacturer’s specifications (Takara Bio, Tokyo, Japan). cDNA (0.4 µg) was used for qRT-PCR. The relative transcript levels of *sufD*, *sufS*, *sufU,* and *sufB* were determined with 16S rDNA as a control by the SYBR Premix Ex Taq (Tli RNaseH Plus) kit (Takara Bio, Tokyo, Japan). Primers for *sufC*, *sufD*, *sufS*, *sufU*, *sufB*, *nifH*, *nifD*, *nifU,* and 16S rDNA used for RT-PCR or qRT-PCR are listed in Appendix A.

### 4.6. Western Blot Assays for NifH and NifD Expression

Cultures of *P. polymyxa* WLY78, ∆*nifS-*like, ∆*sufCDB,* and ∆*yutI* were grown under N_2_-fixing conditions and harvested after 20 h of incubation. Cells were collected and disrupted in lysis buffer (50 mM NaH_2_PO_4_, 300 mM NaCl, and 10 mM imidazole) by sonication on ice. Cell debris were removed by centrifugation, and 40 µg of total proteins was analyzed by SDS-PAGE (10% acrylamide) and immunoblotting. Proteins were blotted onto a nitrocellulose (NC) membrane, and the Fe and MoFe proteins were detected using polyclonal anti-NifH and anti-NifD, respectively. Binding of antibodies was visualizedby enhanced chemiluminescence (ECL) (ComWin Biotech, Beijing, China).

## 5. Conclusions

In *P. polymyxa* WLY78*,* the *sufC*, *sufD,* and *sufB* genes of the *suf (sufCDSUB*) operon, the *nifS*-like gene, and the *yutI* gene are required for the Fe–S cluster biosynthesis of nitrogenase. The cysteine desulfurase NifS-like catalyzes the production of sulfur, and SufCDB proteins provide a molecular scaffold for the assembly of the Fe–S cluster. YutI is an Fe–S carrier that is involved in nitrogen fixation.

## Figures and Tables

**Figure 1 ijms-22-03771-f001:**
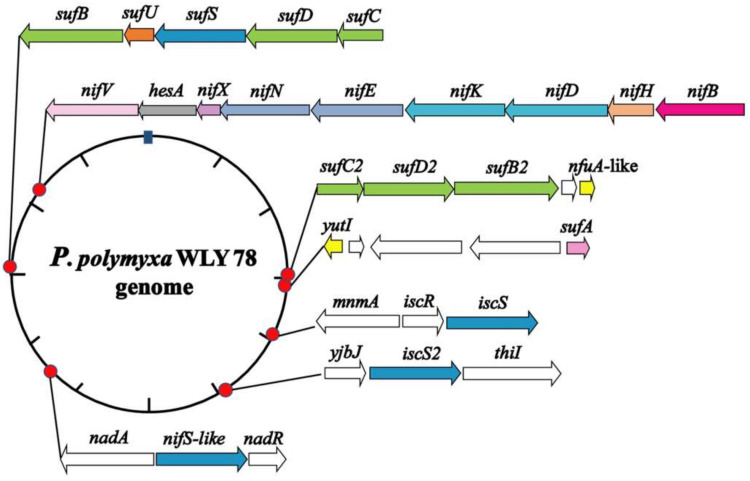
Genomic location of the *Paenibacillus polymyxa* WLY78 *nif* gene cluster and the putative genes involved in Fe–S cluster biogenesis. The schematic diagram representation shows the relative location, size, and orientation of the genes, and relevant flanking genes.

**Figure 2 ijms-22-03771-f002:**
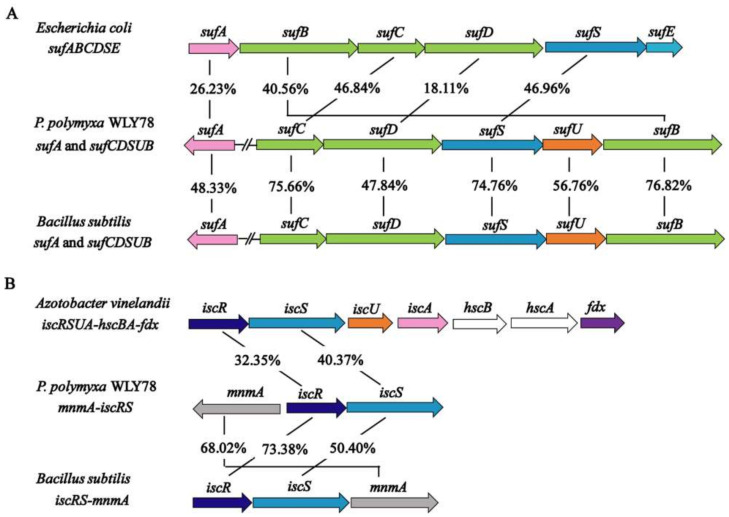
Comparison of the *P. polymyxa* WLY78 *suf* and *isc* cluster with other bacteria. (**A**) Comparison of the *P. polymyxa sufCDSUB* operon and the *sufA* gene with *Escherichia coli* and *Bacillus subtilis suf* operons and *sufA* genes. (**B**) Comparison of the *P. polymyxa iscRS* operon with the *Azotobacter vinelandii* and *B. subtilis isc* operons. For gene products with similar sequences, their amino acid identity (%) is indicated.

**Figure 3 ijms-22-03771-f003:**
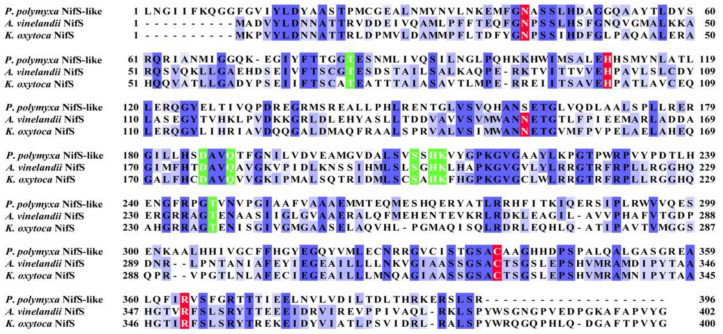
Comparison of amino acid sequences of the NifS-like protein from *P. polymyxa* WLY78 and NifS from *A. vinelandii* and *K. oxytoca*. Amino acid sequences were aligned using the Clustal Omega program (https://www.ebi.ac.uk/Tools/msa/clustalo/, accessed on 25 March 2021). Conserved residues involved in pyridoxal-phosphate (PLP) binding and activity are indicated by a green background. Invariable cysteine and other residues involved in substrate binding are indicated by a red background.

**Figure 4 ijms-22-03771-f004:**
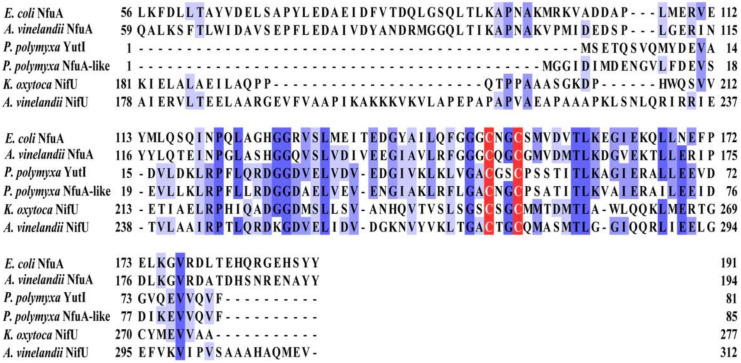
Comparison of NifU-like proteins (YutI and NfuA-like) from *P. polymyxa*, NifU from *A. vinelandii*, NifU from *K. oxytoca*, and NfuA from *E. coli*. Protein sequences were aligned using the Clustal Omega program (https://www.ebi.ac.uk/Tools/msa/clustalo/, accessed on 25 March 2021). Conserved CXXC motifs are highlighted with a red background.

**Figure 5 ijms-22-03771-f005:**
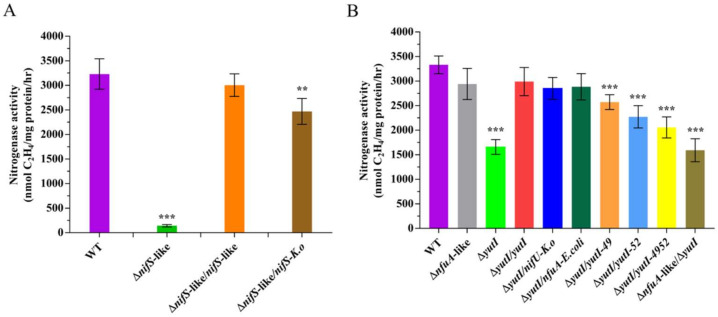
Effect of disruption of *nifS-*like and *nifU-*like genes on nitrogenase activity. (**A**) The nitrogenase activities of wild-type (WT), Δ*nifS-*like (deletion mutant), Δ*nifS-*like*/nifS-*like (complementation strain), and Δ*nifS/nifS-K.o* (complementation by *K. oxytoca nifS*). (**B**) Nitrogenase activities of wild-type (WT), Δ*nfuA* and Δ*yutI* (*nfuA*-like and *yutI* deletion mutant), Δ*yutI/yutI* (*yutI* complementation strain), Δ*yutI/nifU* (complementation by *K. oxytoca nifU*), Δ*yutI/nfuA-E. coli* (complementation by *E. coli nfuA*), Δ*yutI/yutI-49,* Δ*yutI/yutI-52*, Δ*yutI/yutI-4952* (cysteine variants of the YutI complementation strain), and Δ*nfuA*-like/Δ*yutI* (*nfuA*-like and *yutI* double mutant). The nitrogenase activities were assayed by the C_2_H_2_ reduction method and expressed in nmol C_2_H_4_/mg protein/hr. The nitrogenase activity of the WT strain was used as a control. Results are representative of at least three independent experiments. Error bars indicate the SD. ** *p* < 0.01; *** *p* < 0.001.

**Figure 6 ijms-22-03771-f006:**
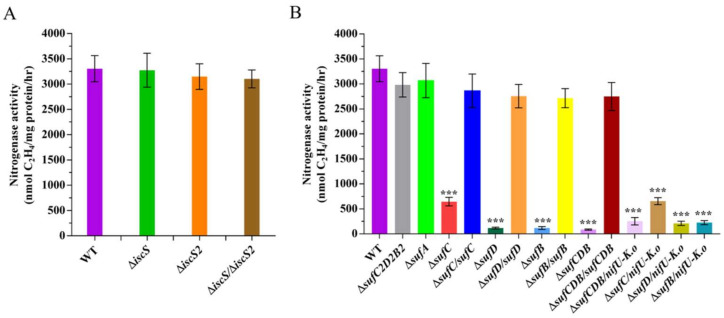
Effect of disruption of *iscS* and *suf* genes on nitrogenase activities. (**A**) Nitrogenase activities of wild-type (WT), Δ*iscS* (deletion mutant), Δ*iscS2* (deletion mutant), and Δ*iscS/*Δ*iscS2* (double deletion mutant). (**B**) Nitrogenase activities of wild-type (WT), Δ*sufC2D2B2* (deletion mutant), Δ*sufA* (deletion mutant), Δ*sufC* (deletion mutant), Δ*sufC/sufC* (complementation strain), Δ*sufD* (deletion mutant), Δ*sufD/sufD* (complementation strain), Δ*sufB* (deletion mutant), Δ*sufB/sufB* (complementation strain), Δ*sufCDB* (deletion mutant), Δ*sufCDB/sufCDB* (complementation strain), and Δ*sufCDB/nifU-K.o*, Δ*sufC/nifU-K.o*, Δ*sufD/nifU-K.o*, Δ*sufB/nifU-K.o* (complementation by *K. oxytoca nifU*). The nitrogenase activities of these strains were assayed by the C_2_H_2_ reduction method and expressed in nmol C_2_H_4_/mg protein/hr. The nitrogenase activity of the WT strain was used as a control. Results are representative of at least three independent experiments. Error bars indicate the SD. *** *p* < 0.001.

**Figure 7 ijms-22-03771-f007:**
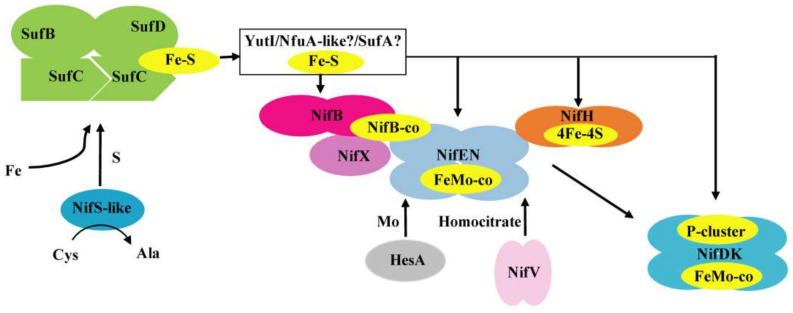
A model of Fe–S cluster biosynthesis of nitrogenase in *P*. *polymyxa* WLY78. The SufBCD complex acts as a scaffold complex receiving sulfur from NifS-like and iron from a still unknown donor. YutI (possibly work with SufA or/and NfuA-like) is as an Fe–S carrier that may receive an Fe–S cluster from the SufBCD scaffold and transfer it to nitrogenase component proteins. The model shows in yellow the Fe–S cluster. The 4Fe–4S cluster is inserted into apo-NifH to activate it. Simple Fe–S clusters were reconstituted on NifB and are converted into NifB-co. NifEN binds NifB-co and converts it into FeMo-co. HesA and NifV provide Mo and homocitrate. NifX may assist NifB, NifEN, and NifDK in FeMo-co precursor trafficking. The FeMo-co and in situ assembly of the P-cluster are inserted into apo-NifDK to form holo-NifDK.

## Data Availability

Not applicable.

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
