# Peer review of "Identification of Genes Involved in Fe–S Cluster Biosynthesis of Nitrogenase in Paenibacillus polymyxa WLY78"

_ijms, 2021, doi:10.3390/ijms22073771_

Round 1
Reviewer 1 Report
The study reported by Li et al. in Paenibacillus polymyxa WLY78 provide a clear description of some of the genes involved in the maturation of the Fe-S clusters of nitrogenase. This work is highly relevant for engineering of artificial nitrogen fixation systems. It also provide new interesting insights into the non-redundancy of some of the Fe-S biogenesis actors and besides the ability of some of them to interchange. However, there are still some missing links to establish the network of genes responsible for the maturation of the clusters of the nitrogenase in this organism.
Major point:
The authors clearly identified SufS, SufU, SufB, SufC and SufB as major actors of the maturation of the nitrogenase clusters. This subset of proteins performs the synthesis of [2Fe2S] clusters that are the precursors of the [4Fe4S] clusters that will be subsequently used for the maturation of the M and P clusters of nitrogenase. In other organisms, A-type scaffolds (IscA, SufA and NfuA) support the synthesis of [4Fe4S] clusters from the [2Fe2S] building block as well as transfer of the [4Fe4S] to target acceptors. Here the authors identified an NfuA-like protein, YutI, that is possibly involved in this process. Surprisingly, neither SufA, nor the other NfuA-like protein seem to be involved. However, in strains deleted for yutI there is still 70% of nitrogenase activity, which suggests that other proteins compensate for the lack of YutI, possibly SufA and/or the other NfuA-like protein. Therefore, the studies of combinations of double and triple mutants of SufA, NfuA and YutI must be conducted to provide a definite answer on the network of proteins involved in the maturation of nitrogenase.
Reviewer 2 Report
The authors have investigated the genetic requirements for nitrogenase directed Fe-S cluster biosynthesis in the Gram-positive bacterium Paenibacillus polymyxa WLY78. P. polymyxa WLY78 has some interesting features for engineering of eukaryotic N2 fixation, such as a minimal nif gene cluster. While several aspects of the manuscript are interesting, this reviewer feels that biochemical evidences for several claims are lacking.
Major comments
- All hard data presented is in the form of in vivo nitrogenase assays. From these data the authors draw conclusions that should be backed up by biochemical data. For example, the authors present a model (Figure 5) in which a working model and a complex is shown. Does this complex exist? That should be possible to test relatively easy in pulldown experiments or similar. Also, Figure 5 is too simplified. Where is NifB and NifEN? No clusters are indicated at holo-NifH. What clusters are synthesized? Nitrogenase have several cluster essential for activity. Can the authors isolate these proteins in order to test these questions?
- It should be shown that in-frame deletions are affecting protein expression levels, to link with reduced nitrogenase activities. I understand that antibodies are not available, but perhaps the authors could perform some mass spec analysis? Also, they should show that expression levels of other genes in the operon is not affected by the single gene deletions (e.g. line 300). The same goes for the complementation studies (show that the proteins are expressed).
- The manuscript would be much stronger if proteins could be isolated for biochemical characterisations. For example to show that they can harbour Fe-S clusters, to enable testing their activities in vitro
- What is the nitrogenase activity of a double ΔyutI/ΔnfuA-like mutant? Do they have complementary functions?
- How well does the prediction tool used for Table S1 work? NifX is not in the list for example (but should be there according to line 70).
Minor comments
- Lines 75 to 78: rephrase, because firstly it is stated that polymyxa nif cluster lacks nifS and nifU, and then, you want to identify whether there are “nifS and nifU” or other Fe-S cluster biosynthetic genes […]. It is meant at other genomic location?
- Line 83 results doesn’t support that YutI is essential for nitrogen fixation as activity of the ΔyutI strain is around 50% of the WT strain. In line 86 to 87, “yutI is required for the maturation of nitrogenase” is over interpreting the data, yutI has a detrimental effect on nitrogenase activity, but it is not demonstrated that it is directly related to maturation of nitrogenase. For this, purification of nitrogenase components should be performed and Fe-S cluster integrity and occupancy analyzed. Rephrase.
- Line 88: “maturation of nitrogenase”, there is no analysis of nitrogenase integrity, nitrogenase cofactors or precursors. The provided data does not support this statement.
- Line 104 what does “tightly arranged” mean?
- Line 110 and elsewhere: WLY78 is sometimes written out, sometimes omitted. It should be consistent throughout the manuscript.
- Language and grammar should be improved. For example line 20 (specific), 62 (example), 113 (rephrase), 213 (were), 220 (similar activity to), 226 (genes), 235 (do not have), 236 (their primary sequences), 245 (linked?), 257 (specific?), 286 (suggesting), 288 (participates), 294 (lethal for subtilis), and more.
- Line 124: what is meant with invariable residues Cys? Conserved?
- Line 146 “nitrogen limiting conditions”
- Line 160 C2H2 reduction method.
- Line 167: should be “in Azotobacter vinelandii”?
- Line 171: 63% activity. Is this correct? It looks like less than 50% in the Figure 3B.
- Line 179. yutI mutant should be: ΔyutI
- Lines 189-192: Makes little sense. The authors should carefully read through the manuscript, also repeated words at line 416.
- Line 205: C2H2 reduction assay.
- Lines 210, 211, 224 and throughout the manuscript: The authors should make sure genes are in italics, or if proteins are meant in upper case.
- Line 212: This statement should be tested/shown using ΔsufCDB/ΔsufC2D2B2 a double mutant.
- Line 217: comparatively analysed?
- Lines 223-225: Any suf mutant? Only the triple mutant complemented in Figure 4B. Is NifU expressed? Easy to confirm by WB.
- Lines 225 to 227: sufCBD is specifically involved in the Fe-S cluster assembly of nitrogenase in Paenibacillus is true. But the reason why ΔSufCDB phenotype cannot be complemented with Klebsiella’s NifU could be because of its sulfur donor NifS is not present. SufSU may not be able to load sulfur atoms into NifU and therefore result in low nitrogenase activity. That doesn’t mean that the Nif system is not compatible with Paenibacillus Nif proteins. Summarizing, the first part of the sentence is not necessarily connected to the conclusion drawn in the second part of the sentence.
- Line 226: not shown specifically
- Line 229: Nitrogenase is a complex metalloenzyme.
- Line 235: what does “characteristics” mean in this sentence? Rephrase.
- Lines 237: almost certainly responsible? It does not sound very scientific, please rephrase.
- Line 253: Add also Jacobson et al. 1989, which is the original paper about ΔnifS mutant in Azotobacter.
- Line 261: ref 38 refers only to pylori, not several bacterial species.
- Line 267: “essential” in P. polymyxa. Essential for what physiological process? Mutation of ΔyutI is not deletereus, so this protein is not essential for P. polymyxa and nitrogenase activity is not completely abolished in the mutant, so yutI is not essential.
- Line 291: Ref 32 correct?
- Figure 1: I would recommend to indicate the genomic location of the nif cluster as well.
- Figure 4: ΔsufA not described in the legend.
Reviewer 3 Report
The paper is very interesting.
I suggest the minor modification concerning the following statement:
Sequence analysis indicates the presence of two genes (nfuA-like and yutI genes), encoding for 85 and 81 amino acids respectively, that share similarity with the C-terminus
of the Fe-S scaffold protein NifU and Fe-S carrier NfuA in P. polymyxa WLY78 (Figure 2).
Why don't you provide a figure with the comparison of the two sequences?
In my opinion, this figure will help to better understand the discussion (line 248).
Round 2
Reviewer 1 Report
The authors have partially addressed the question raised by the reviewer. Data are presented on a double mutant NfuA, YutI, which does not exacerbate the phenotype, thus indicating that NfuA does not compensate for the loss of YutI. Other factors can thus work with YutI, possibly SufA even if a SufA mutant does not have a phenotype. A final answer to this question requires extra hard work to generate double and triple mutants.
For this manuscript, authors should add YutI and as yet undefined factors including SufA and NfuA with question marks on Fig.7
Author Response
Reviewer 1:
Comments and Suggestions for Authors
The authors have partially addressed the question raised by the reviewer. Data are presented on a double mutant NfuA, YutI, which does not exacerbate the phenotype, thus indicating that NfuA does not compensate for the loss of YutI. Other factors can thus work with YutI, possibly SufA even if a SufA mutant does not have a phenotype. A final answer to this question requires extra hard work to generate double and triple mutants. For this manuscript, authors should add YutI and as yet undefined factors including SufA and NfuA with question marks on Fig.7
Response: Thanks for your comments for the manuscript. We have added YutI and as yet undefined factors including SufA and NfuA with question marks on Fig.7
Reviewer 2 Report
Major comments (numbering according to original points)
- I understand that purification from the native host is a difficult task. Still, there is not a single biochemical evidence in this manuscript to support the title (“the roles of the distinct iron-sulfur cluster assembly systems…”). This title is overselling the data provided in the manuscript. As the authors say, they are now expressing and purifying these Fe-S proteins from E. coli. At least the authors should prove that this complex is formed. Simple UV-vis analysis could determine if and what type of cluster is associated to the Suf complex.
- The provided WB for HDK protein expression (S3) is not quantitative as there is no loading control provided. The authors claim that 40 ug was loaded, but these small differences could be from errors in the protein quantifications. A house keeping protein should be included, or minimally the Ponceau stained membrane or a corresponding Coomassie gel. Also, if they want to draw any quantitative conclusion there should be independent experiments confirming these small effects. Also, the qPCR in S2 makes little sense. In panel A all other genes are included, but in B only sufB. What is the reason for this, were the other genes affected? All genes should be tested.
- I understand the authors are planning a follow-up to this study. Still, I still believe my comment above is valid that some initial biochemical analysis could be provided here, at least to show that the purified E. coli proteins harbour Fe-S clusters.
Minor comment (numbering according to original points)
Lines 223-225 in the first version. NifU antibodies should be possible to acquire (or they can express a epitope-tagged NifU protein), or at least include data that the nifU gene is expressed.
Other comments
The language used in the manuscript is still not good, and I strongly suggest that an English native speaker looks at it. For example, line 79 “there are”, line 238 “Effect Nitrogenase activities”, line 249 and 345 “Essentially Required” (what does this mean?), etc.
Round 3
Reviewer 2 Report
I think the authors have no responded to my most urgent questions and the manuscript is suitable for acceptance.